# Genome-Wide Meta-Analysis of QTLs Associated with Root Traits and Implications for Maize Breeding

**DOI:** 10.3390/ijms24076135

**Published:** 2023-03-24

**Authors:** Krishna Sai Karnatam, Gautam Chhabra, Dinesh Kumar Saini, Rajveer Singh, Gurwinder Kaur, Umesh Preethi Praba, Pankaj Kumar, Simran Goyal, Priti Sharma, Rumesh Ranjan, Surinder K. Sandhu, Ramesh Kumar, Yogesh Vikal

**Affiliations:** 1School of Agricultural Biotechnology, Punjab Agricultural University, Ludhiana 141001, India; 2Department of Plant Breeding and Genetics, Punjab Agricultural University, Ludhiana 141001, India; 3Indian Institute of Maize Research, Ludhiana 141001, India

**Keywords:** abiotic stress, candidate genes, GWAS, expression analysis, meta-QTLs, root systemarchitecture

## Abstract

Root system architecture (RSA), also known as root morphology, is critical in plant acquisition of soil resources, plant growth, and yield formation. Many QTLs associated with RSA or root traits in maize have been identified using several bi-parental populations, particularly in response to various environmental factors. In the present study, a meta-analysis of QTLs associated with root traits was performed in maize using 917 QTLs retrieved from 43 mapping studies published from 1998 to 2020. A total of 631 QTLs were projected onto a consensus map involving 19,714 markers, which led to the prediction of 68 meta-QTLs (MQTLs). Among these 68 MQTLs, 36 MQTLs were validated with the marker-trait associations available from previous genome-wide association studies for root traits. The use of comparative genomics approaches revealed several gene models conserved among the maize, sorghum, and rice genomes. Among the conserved genomic regions, the ortho-MQTL analysis uncovered 20 maize MQTLs syntenic to 27 rice MQTLs for root traits. Functional analysis of some high-confidence MQTL regions revealed 442 gene models, which were then subjected to in silico expression analysis, yielding 235 gene models with significant expression in various tissues. Furthermore, 16 known genes viz., *DXS2*, *PHT*, *RTP1*, *TUA4*, *YUC3*, *YUC6*, *RTCS1*, *NSA1*, *EIN2*, *NHX1*, *CPPS4*, *BIGE1*, *RCP1*, *SKUS13*, *YUC5*, and *AW330564* associated with various root traits were present within or near the MQTL regions. These results could aid in QTL cloning and pyramiding in developing new maize varieties with specific root architecture for proper plant growth and development under optimum and abiotic stress conditions.

## 1. Introduction

The root system architecture (RSA) plays a crucial role in plant growth and stress responses. It is the biosynthetic site of phytohormones required for plant development [1,2,3,4]. The RSA refers to the spatial arrangement of the root system or the placement of root axes. Overall, the potential growth and productivity of crop plants depend on the physiology and architecture of the root system or root ideotype to a great extent [5]. Maize (*Zea mays* L.) has sophisticated RSA, and its genome presents exceptional diversity for the root traits [6]. It is the second most widely grown cereal crop after wheat, covering 197 million hectares of land globally and producing 1137 million tons [7]. On genetic and physiological grounds, maize has been proven as an excellent model for exploring the relationships among root structure, genetic variation, and gene regulation, and their interaction with the rhizosphere microbiota [8]. Apart from the C4 pathway, the unique root architecture of maize facilitates the easy absorption of nutrients and stress tolerance compared to wheat and rice [9].

The maize root system is made up of different root types that form and function at different developmental stages (Figure 1) [10,11]. The RSA of maize consists of both embryonic and post-embryonic roots [12]. The embryonic root system consists of primary, seminal, and crown roots, while the post-embryonic root system consists of lateral and brace roots and emerges a few weeks after germination [13]. The crown roots formed at the underground nodes are called crown roots, while those at the above-ground nodes are called brace roots [14]. Seminal and primary roots are axile roots, while those arising from axile roots are termed lateral roots (LRs). LRs are 15–35 times the length of axile roots, allowing for better water and mineral absorption in later stages. Because these roots contain approximately 75% of the large xylem conduits between the root system and the stem, the nodal roots (crown and brace root) from the two uppermost tiers are considered crucial for nutrient and water uptake [15]. The genetics of root traits is complex and is controlled by multiple genes [16].

A decrease in cost for genotyping and high precision in phenotyping using modern approaches has led to the identification of several QTLs/genes associated with different traits in maize [17]. Identifying QTLs and markers linked to RSA is essential for exploiting root traits in maize breeding programs through marker-assisted selection [18]. Several QTLs/genes associated with different root traits have been identified in maize over the last few decades [19]. However, the use of these QTLs for breeding remains a concern owing to the differences in genetic backgrounds, mapping population type and size, and statistical methods for QTL mapping across the studies. Validation and fine mapping of QTLs and genes could be used to verify and narrow down the genomic regions governing target traits, but these are tedious and time-consuming approaches. Genome-wide association study (GWAS) is another powerful tool that has been employed successfully to identify significant markers associated with different traits in maize and other cereals [20,21].

The meta-QTL (MQTL) analysis is a powerful method for identifying consensus regions by combining QTL information from multiple studies [22]. Several recent MQTL studies have been conducted on different crops [23,24,25,26,27,28,29,30]. In maize, MQTL analysis has been conducted for different traits, including multiple abiotic stress tolerance [31], disease resistance [32], root-related traits [33], and popping traits [34]. Besides maize, MQTL studies have been reported in other crops for root traits, including rice [35,36] and wheat [26,37]. Earlier MQTL analysis for root traits involved QTLs reported from only 20 published interval mapping studies using 512 QTLs [33]. The present study involves MQTL analysis based on 43 interval mapping studies and identifies the potential candidate genes (CGs) for root-related traits. Overall, the objectives of the present study were as follows: (i) to identify the consensus regions governing the maize root traits by meta-analysis, (ii) to validate the MQTLs with GWAS results, (iii) to study the synteny and collinearity among maize, sorghum, and rice genomes and identify the ortho-MQTLs between maize and rice, (iv) to identify the promising CGs within the selected MQTL regions.

## 2. Results

### 2.1. Characterization of QTLs Associated with Root Traits

In total, 917 QTLs from 43 QTL mapping studies on different root traits during 1998-2020 were collected (Appendix A). The mapping populations in these studies consisted of RILs, F2/F2:3 lines, backcross populations, and doubled haploid populations of sizes ranging from 75 to 586 lines (Figure 2a). Four studies used teosinte-derived maize (subspecies *nicaraguensis*, *huehuetenangensis*, and *luxurians*) populations. The LOD (logarithm of odd) score of these QTLs ranged from 1.5 to 14.66, with an average of 3.86 (Figure 2b). The phenotypic variation (PVE) explained by individual QTLs ranged from 0.5 to 49.9%, with an average of 10.2% (Figure 2c). The distribution of the number of QTL-associated root traits varied significantly among the different maize chromosomes, with a minimum of 73 QTLs located on chromosome 10 to a maximum of 133 QTLs located on chromosome 1. Most QTL mapping studies utilized simple sequence repeats (SSR) as markers and composite interval mapping methods. Further, the number of QTLs available for different traits differed, with the lowest number of QTLs (2) available for root cell percentage and the maximum number of QTLs (295) for root length (Figure 3a).

### 2.2. Consensus Map, Projected QTLs and MQTLs for Root Traits

The consensus map comprised 19,714 markers with a total map length of 7898.35 cM. The map length ranged from 575.73 (*Zm*10) to 1310.63 cM (*Zm*1), while the number of markers localized on individual chromosomes ranged from 1359 (*Zm*9) to 3167 (*Zm*1). The marker densities among maize chromosomes ranged from 1.96 (*Zm*9) to 3.02 markers per cM (*Zm*2). Markers on individual chromosomes were not evenly distributed, with two ends of the chromosomes carrying markers of varying densities. Out of 917 QTLs, a total of 631 QTLs were projected onto the high-density consensus map; 286 QTLs were not projected either because of (i) the absence of common flanking markers in the consensus map or (ii) the large confidence interval of the QTLs.

The number of projected QTLs among different chromosomes also varied; the lowest projected QTLs (15) were on *Zm*8, and the maximum projected QTLs (92) were on *Zm*1 with an average of 63 QTLs per chromosome (Figure 3a). Among the 15 major root traits, a minimum of only one projected QTL was associated with root cell percentage, and a maximum number of projected QTLs (204) were associated with root length (Figure 3b; Appendix A). Meta-analysis of 631 QTLs led to the identification of 68 MQTLs and 6 singletons or single QTLs associated with different root traits (Figure 4; Table 1). Among the 68 MQTLs, 9 MQTLs were considered breeder’s MQTLs (bMQTLs), and 21 MQTLs as high-confidence MQTLs (hcMQTLs).

### 2.3. GWAS Validated MQTLs

The physical coordinates of MQTLs associated with root traits identified in this study were compared with genomic positions of marker-trait associations (MTAs) related with root traits previously reported in GWA studies in maize (Appendix A). Among the 68 MQTLs, as many as 36 MQTLs (52.9 % of the total) were co-localized with at least one MTA (Figure 4). Overall, as many as 95 MTAs were co-localized with 36 MQTLs. Some MQTLs were co-localized with more than one MTA (for instance, MQTL2.1, 2.2, and 7.7 each co-localized with more than 9 MTAs identified in different GWAS). Among the 36 GWAS-validated MQTLs, as many as 16 MQTLs (MQTL 1.1, 1.2, 2.1, 2.2, 2.4, 3.1, 3.5, 4.2, 5.6, 6.1, 6.3, 6.4, 7.1, 7.2, 9.1 and 10.5) included at least 10 projected QTLs from different studies (Appendix A). The following traits associated with the MTAs were used to validate the MQTLs: crown root angle, brace root angle, brace root number, crown root number, primary root diameter, crown root diameter, total root number, root area, primary root length, total root length, seminal root length, lateral root length, root surface area, root shoot ratio, root calcium content, number of root tips, and root dry weight (Appendix A).

### 2.4. Candidate Genes Available from hcMQTL Regions and Their Expression Analysis

Prediction of genes within hcMQTL regions led to the identification of 442 gene models (Appendix A). Gene Ontology (GO) of 442 gene models revealed several GO terms, out of which some of the key and most abundant GO terms include those involved in biological processes such as protein–DNA complex assembly, nucleosome organization, chromatin assembly, nucleosome assembly, chromatin assembly or disassembly, DNA packaging, DNA conformation change, macromolecular complex assembly, etc. Similarly, important GO terms in the molecular functions category included those involved in NAD or NADH binding, molecular transducer activity, ribonuclease activity, signal transducer activity, etc. (Appendix A). Some of the genes with similar functions were found in the maize MQTL regions, which included the following: 13 genes for AP2/ERF domain-containing proteins, 13 for protein kinases, 9 for zinc finger proteins, 7 for glycoside hydrolases, and 5 for exostosin-like proteins. In some MQTL regions, gene clusters of specific gene superfamilies were also found, for instance, (i) protein kinase domain superfamily, (ii) methyltransferase superfamily, and (iii) AP2/ERFTF (Appendix A).

The expression data (transcript profiling) for gene models were retrieved from “qteller” of maizeGDB. Abundance or higher expression of a particular gene at different time points in different tissues have been used to extract only those genes that are significantly upregulated related to root traits. Among these 442 gene models, 235 were significantly expressed (each with >3 TPM expression) in different tissues. Most highly expressed genes encode for the following proteins: (i) transcription factors, (ii) proteins involved in ROS activity, (iii) casparian strip protein domain containing proteins, (iv) protein kinases, and (v) protein channels for ions transfer. Based on the available literature, functional relevance was examined for the proteins encoded by 235 genes. This analysis identified 69 genes that encode proteins functionally essential for RSA. Therefore, these genes were recommended as potential candidate genes for the corresponding hcMQTL regions (Appendix A). The heat map of the selected CGs from different root tissues is represented in Appendix A.

### 2.5. Conserved Genomic Regions and Ortho-MQTLs among the Cereals

The conserved regions among maize, rice and sorghum genomes (identified based on the gene models available from maize MQTL regions) are shown using a circos plot (Figure 5). Maize had 2366 syntenic genes with sorghum and 1199 orthologous genes with rice. The root syntenic genes in sorghum ranged from 40 (*Sb*4) to 451 (*Sb*10), whereas in rice, it varied from 32 (*Os2*) to 245 (*Os3*). The substantial search for gene models of rice and maize root MQTLs identified 20 maize MQTLs syntenic to 27 rice MQTLs (Figure 5; Appendix A). One maize MQTL was found to be syntenic to two rice MQTLs, and another MQTL was syntenic to three rice root MQTLs. Maize MQTL4.7 was syntenic to two rice MQTLs located on chromosomes 2 and 11 (viz., MQTL2.3 and MQTL11.3), and MQTL7.7 showed synteny to three rice MQTLs (MQTL7.3, MQTL9.1, and MQTL9.2) located on chromosomes 7 and 9. Further, each of nine ortho-MQTLs detected between maize and rice shared a minimum of ten conserved gene models at corresponding positions in maize and rice genomes (Figure 6). The ortho-MQTL analysis could not be performed in sorghum, as no MQTL analysis has been published for root traits in sorghum.

### 2.6. Adjacent MQTLs with Known Characterized Maize Root Genes and Orthologs of Rice Genes in MQTL Regions

Sixteen known maize genes associated with root traits were detected within or near the MQTL regions. Six genes, *DXS2*, *PHT*, *RTP1*, *TUA4*, *YUC3*, and *YUC6*, were found to be localized within MQTL7.5, 5.10, 4.6, 5.7, 8.2, and 4.7 (Table 2). Ten genes, including *RTCS1*, *NSA1*, *EIN2*, *NHX1*, *CPPS4*, *BIGE1*, *RCP1*, *SKUS13*, *YUC5*, and *AW330564*, were identified within ±1 Mb of MQTL genomic regions (Table 3). Further, orthologs of eighteen rice genes (*OsACS7*, *OsAMT1*, *OsCDPK10*, *OsCIPK14*, *OsCIPK15*, *OsCIPK23*, *OsCKX1*, *OsEXPA10*, *OsHAK10*, *OsLAC17*, *OsMGT*, *OsNAC7*, *OsWRKY68*, *OsYUCCA10*, *OsYUCCA12*, *OsYUCCA13*, *OsYUCCA14*, *OsYUCCA9*, and *OsDREB1E*) associated with different root traits were also identified in the 14 maize MQTL regions (Table 3).

## 3. Discussion

Roots play a significant role in plant establishment, nutrient and water uptake, and abiotic stress responses for increased yield. Maize should have a better root idiotype for its function to increase productivity. As a result, to develop a maize seedling with early vigour, a breeder should concentrate on primary and seminal root traits. Similarly, in the vegetative stage, maize breeders should focus on lateral and crown roots to improve nutrient uptake and lodging resistance. Root angle, length, number, diameter, enzyme assay (associated with phosphorus uptake), and weight are the primary root traits, whereas root area, volume, density, and strength are considered correlated or related traits [66]. In the present study, we also focussed on root enzymes such as acid phosphatase (APS) and H^+^ secretion from roots that have a role in phosphorous uptake. Root traits are highly quantitative, and combining different maize root traits for better architecture is difficult.

Scientific communities from all over the world have reported several QTL mapping studies for RSA in maize. However, using all of that data for breeding programs remains a question. For this purpose, a meta-analysis can be used to identify MQTLs responsible for different maize root traits. Earlier, Guo et al. [33] conducted a meta-QTL analysis of maize root architecture and reported 53 MQTLs with 428 projected QTLs for 23 maize root traits. Nevertheless, the results of the MQTL significantly correlated with the number of initial QTLs. Many maize root QTLs are regularly identified and published [19,21,67]; thus, it is critical to integrate new QTL data to identify more stable and consistent MQTLs. Thus, we performed a meta-analysis of 917 QTLs and identified 68 MQTL regions responsible for important root traits. Several QTLs identified for root behavioural traits under abiotic stresses such as water logging, drought, and different levels of nutrient fertilization were also considered for meta-analysis. Unlike the study by Guo et al. [33], the current work targeted 75 root traits and reported 185 more projected QTLs in the MQTLs regions. Further, 22 MQTLs overlapped with 23 MQTLs reported in an earlier study [33] based on the physical coordinates of MQTLs. Since different genome assemblies were used to extract the physical coordinates of MQTLs, this comparison might not be entirely accurate; as the most recent reference genome assembly (*Zm*-B73-REFERENCE-NAM-5.0) was used in the current study as opposed to the old reference genome assembly (Maize_B73_RefGen_v4) used previously [33]. Additionally, compared to Guo et al. [33], more than twice the QTLs and mapping studies were considered in the current study. This could be related to the precise refinement of the physical coordinates of the MQTLs predicted.

In this study, the majority of MQTLs were found to be concentrated at the fore ends of chromosomes, which are known to be regions of high gene density (https://www.maizegdb.org/genome/assembly/Zm-B73-REFERENCE-NAM-5.0 (accessed on 14 June 2022)) [68]. This pattern of MQTL distribution has also been observed in several earlier meta-analyses of different crops, including maize [27,30,32,69]. There are two plausible explanations for this pattern of MQTL distribution on chromosomes. Firstly, the higher density of markers at the fore ends of the chromosomes may have contributed to the detection of more MQTLs in these regions. This was likely due to the inclusion of markers with different types and numbers in the independent genetic map used for constructing the consensus map. Taking into account the physical positions of markers, MQTLs were found to be scattered throughout almost the entire span of the chromosomes, consistent with the physical positions of MQTLs identified in previous meta-analysis studies. Differences in the physical positions of MQTLs reported in earlier studies [33,70] and the present study may be attributable to the usage of different genome assemblies for extracting the physical positions of MQTLs. The current study implemented the latest reference genome assembly (*Zm*-B73-REFERENCE-NAM-5.0), whereas earlier meta-analysis studies employed the earlier reference genome assembly (Maize_B73_RefGen_v4) [33,71]. Secondly, although some initial QTLs were situated at the lower ends of the genetic maps of the chromosomes, these QTLs were not successfully projected, potentially due to a lack of common markers between the initial genetic maps and the consensus map. Consequently, this may have resulted in the absence of MQTLs on the lower ends of the chromosomes. The maximum number of QTLs was projected on chromosome 1 with 92 QTLs, followed by chromosomes 2, 3 (each with 85) and 5 (83 QTLs), whereas chromosome 8 had the lowest number of QTLs (23). Soriano and Alvaro [37] subcategorized 75 wheat root traits into 15 to better understand the results. Some of the studies used the teosinte of three different sub-species to evaluate water logging tolerance, which is a unique feature of this study.

Among the 68 MQTLs, 9 MQTLs were the most stable and robust, with a minimum of 15 projected QTLs derived from different studies. For instance, MQTL1.1 involved 42 projected QTLs, followed by MQTL10.3 with 28 QTLs. Among the total MQTLs, 18 MQTLs (26.47% of the total) had the projected QTLs of six major root traits (angle, length, number, diameter, enzyme assay, and weight). Furthermore, nine MQTLs were classified as bMQTLs because they had a mean PVE greater than 10%, CI less than 2cM, and involved at least five initial QTLs. Markers flanking these bMQTL should be deployed for marker-assisted selection (MAS) programs for the genetic enhancement of root traits.

GWAS is a powerful tool used to dissect the genetic architecture of complex traits using association panels and permits high-resolution mapping [71]. Low-cost and high-throughput sequencing technologies have enabled us to identify markers significantly associated with various biotic, abiotic, root and quality traits [72]. The MQTLs co-localized with GWAS-based MTAs can be considered more stable and consistent. In the present study, 52.9% of MQTLs (36 out of 68) were validated using GWAS-MTAs obtained from ten maize root GWA studies. Such a comparison of the MQTLs with GWAS-based MTAs has been performed in a few earlier studies in maize [28,32,35,36]. These 36 MQTLs co-localized with MTAs associated with different traits, including root angle, diameter, length, number, area, and root weight. Several significant CGs near MTAs that co-localize with MQTLs have also been identified. These CGs mainly encoded for the following proteins: UDP-glycosyltransferase, serine/threonine-protein kinase, trigger factor-like protein TIG chloroplast, reticulon-like protein, probable cysteine protease, 60 S ribosomal protein, RING/U-box superfamily protein, cytochrome P450 protein, thioesterase superfamily protein, translin, peroxidise, and beta-galactosidase. Mostly, these gene families coincided with the CGs mined within the hcMQTL regions in the present study. These MQTLs are promising and could be used for effective MAS for maize root traits [73,74]. 

The CGs available from MQTL regions are the targets for allele mining [75]. In an earlier study in maize, 1910 gene models were reported within MQTL regions associated with fungal disease resistance, which contained R genes, pathogenesis-related genes, and disease-responsive genes [32]. During the present study, within the selected hcMQTL regions of maize, 442 gene models were identified, with 357 containing known functional protein domains, and the remaining 85 containing unknown protein domains. In total, 69 high-confidence CGs were short-listed based on transcript abundance and functions related to root architecture that can be commenced for future studies. A literature survey revealed the involvement of these 69 genes in cell elongation, division, metabolism, production of ROS enzymes, signalling mechanism, transcriptional and translational control of various other genes, etc.

Casparian strip membrane proteins (e.g., *Zm00001eb166140*) are involved in the apoplastic movement of water to endodermal cells [76]. Zn/Fe permease (e.g., *Zm00001eb213530*) facilitates Zn uptake [77], while alternative oxidases (e.g., *Zm00001eb066910* and *Zm00001eb066920*) are the conserving terminal oxidases of the mitochondrial electron transport chain, which are known to minimize the ROS [78]. Protein kinases (e.g., *Zm00001eb299910* and *Zm00001eb406950*) regulate root hair development by modulating intercellular signal communications [38]. Likewise, zinc fingers (e.g., *Zm00001eb213300*) are also known to promote root hair growth [39]. WRKY transcription factors (e.g., *Zm00001eb071590*) modulate root growth under environmental stresses [40].

In the present study, 20 ortho-MQTLs were conserved between maize and rice genomes. Among the 20 ortho-MQTLs, 7 MQTLs (viz., MQTL 2.2, 2.5, 4.5, 4.7, 7.2, and 7.7) were verified by the rice root meta-QTL studies [35,36]. Such conserved ortho-MQTLs suggest the presence of some common regulatory elements that govern the functions of many root-related genes [35,42]. These syntenic regions consist of many untapped genes in maize which can be targeted and utilized for functional characterization in future studies. The genes present within ortho-MQTL may be used to understand the network of genes involved in cereal root development. The ortho-MQTL analysis also paved the way to understanding the molecular and evolutionary basis of root development in different cereals. Furthermore, markers can be designed from the conserved regions for use in cereal breeding programs for root traits [27].

Several root genes have been cloned and are functionally characterized in different cereals, including maize. Sixteen known maize genes were identified within or near the fifteen MQTL regions. Six genes were identified within the MQTL regions: (i) *pht1* gene localized in MQTL5.1 encodes for phosphate permease involved in phosphate ion uptake from soil [41]; (ii) *YUC3* and *YUC6* co-localized in MQTL4.7 and MQTL8.2, respectively, encode for flavin monooxygenase, which is known to play a role in auxin biosynthesis in maize roots [43]; (iii) *DXS2* encodes deoxy xylulose synthase 2, but its molecular function is unknown, although it is highly expressed in the primary roots of maize seedlings [48]; (iv) *root preferential 1*, *RTP1*, gene localized in MQTL4.6 has also been reported to be highly expressed in roots and involved in suberin pathway [50]; (v) *TUA4* identified in MQTL5.7 encodes for alpha-tubulin, having a role in structural maintenance and cell division in root cells [44].

The remaining ten genes detected near the MQTL regions: (i) *rootless concerning crown and seminal roots1* (*RTCS1*) found near MQTL1.2 play a role in promoting primary root development in maize [46]; (ii) mutants of *big embryo 1* gene (*BIGE1*) detected near MQTL4.3 have been shown to be associated with accelerated maize root initiation as well as enlargement of the embryo scutellum [47]; (iii) *skewed root growth similar 13* (*SKUS13*) gene, identified near MQTL7.1, is involved in directional root growth processes in maize [49]; (iv) genes *NSA1* and *NHX1* detected near MQTL2.4 and MQTL4.2, respectively, encode for protein channels involved in Na^+^ ion effluxes in roots under salinity [45,51]; (v) *CPPS4* residing near MQTL4.3 encodes for copalyl diphosphate synthase4, which is moderately expressed in roots when exposed to abiotic stress [52]; (vi) *root cap protein 1* (*RCP1*) identified near MQTL6.5 is highly expressed in the outer most root cap cells and is involved in root development [54]; (vii) *YUC5*, localized near MQTL7.5, is known to be involved in auxin biosynthesis [53]; (viii) *ethylene insensitive 2* (*EIN2*), present near MQTL3.1, is known to be involved in the synthesis of metabolites in roots under stress [61]; (ix) gene *AW330564*, identified near MQTL10.1, encodes for a root phototropism-like protein which is known to regulate positive root phototrophism [59]. These known genes were spread across the entire genome, verifying the efficacy of MQTLs identified on different chromosomes and demonstrating that multiple regions of the maize genome were involved in the growth and development of different maize root types (Table 2).

During the present study, several maize orthologs of rice genes were also identified within some MQTL regions (Table 3). These genes belong to five categories: (i) Genes involved in nutrient uptake: *OsAMT1* (ammonium transporter; maize ortholog-*Zm00001eb077350*), detected in MQTL2.6, is known to regulate ammonium uptake in roots [64]. *OsACS7* (ACC synthase; *Zm00001eb308860*), localized in MQTL7.7, is known to be involved in phosphate uptake in rice [57]. MQTL6.7 carried orthologs of two rice genes, *OsMGT* (magnesium transporter; *Zm00001eb274990*) and *OsHAK10* (potassium transporter; *Zm00001eb275630*) that are known to be involved in the uptake of magnesium ions from the soil solution [62] and participated in high-affinity K+ transport [63], respectively. (ii) Genes involved in hormone metabolism: Maize orthologs of six rice genes, namely, *OsYUCCA9*, *OsYUCCA10* (*Zm00001eb332870*), *OsYUCCA12*, *OsYUCCA13*, *OsYUCCA14* (*Zm00001eb168680*), and *OsCKX1* (*Zm00001eb121500*), detected within MQTL3.6, MQTL4.7, and MQTL8.2 are involved in hormone metabolism. *YUCCA* family genes are known to regulate auxin biosynthesis in the root apex transition zone required for root growth and development in maize and rice [79]. *OsCKX1* is involved in cytokinin metabolism and plays a positive role in crown root formation in rice and *Arabidopsis* [55]. (iii) Genes encoding for TFs: The maize orthologs of three rice genes *OsWRKY* (*Zm00001eb071590*), *OSDREB1E* (*Zm00001eb073550*), and *OsNAC7* (*Zm00001eb269810*) (encoding for WRKY, DREB1, and NAC TFs, respectively) were detected within the MQTL2.4, MQTL2.5, and MQTL6.5 regions. *OsNAC*, *OsWRKY*, and *OSDREB1E* family genes were found to be upregulated in rice roots and to modulate root growth and nutrient uptake under stress [56,58,60]. (iv) Genes involved in signalling: The maize orthologs of four rice genes *OsCDPK10* (*Zm00001eb213280*) *OsCIPK14*, *15* (*Zm00001eb406220*), and *OsCIPK23* (*Zm00001eb300010*), encoding for calcium-dependent protein kinases (CDPKs) and calcineurin B-like proteins interacting protein kinases (CIPKs), respectively, were identified in MQTL5.4, MQTL7.2, MQTL10.2 and MQTL10.3 regions. CDPK genes are vital components of stress signalling pathways in rice and *Arabidopsis* roots and other tissues [65]. CIPKs are known to regulate root meristem development by modulating auxin and cytokinin activities [80]. (v) Genes involved in cell elongation and lignin synthesis regulation: The maize orthologs of two rice genes, *OsEXPA10* and *OsLAC17*, encode expansin and laccase proteins, and were identified within the MQTL2.5 and MQTL4.1 regions, respectively. These genes regulate cell elongation, cell division, and lignin synthesis in rice [81,82].

## 4. Materials and Methods

The steps followed for the meta-analysis during the present study are shown in Figure 7 and are summarized as below:

### 4.1. Bibliographic Search for QTL Data Collection

An extensive bibliographic search was performed using different web repositories such as Google Scholar (https://scholar.google.com/ (accessed on 12 March 2022)) and PubMed Central (https://www.ncbi.nlm.nih.gov/pmc/ (accessed on 15 March 2022)) for QTL studies on root traits in maize published up to the year 2020 (Appendix A). The data were retrieved from these studies on the following aspects: (i) QTL name (if provided), (ii) associated root traits, (iii) chromosome number, (iv) LOD score of individual QTLs, (v) phenotypic variation of individual QTLs (R^2^ or PVE), (vi) size and type of the mapping population utilized for mapping, (vii) the peak position of the QTLs, (viii) flanking markers of the individual QTLs and their genetic positions (confidence interval, CI (95%)), (ix) mapping function used to construct the linkage maps. In some studies, the midpoint of the flanking markers served as the QTL’s peak value because the peak position was unavailable. When information on the CI of particular QTLs was lacking, the CIs (95%) were calculated using population-specific formulas [25,83,84] as given below: CI (95%, for DH population) = 287/(PVE × N)
CI (95%, for RIL population) = 163/(PVE × N)
CI (95%, for F_2_ population) = 530/(PVE × N)
where “N” indicates the population size used in the individual mapping studies, and “PVE” is the phenotypic variation governed by each QTL. Data on 75 different root traits were grouped into 15 categories for meta-analysis (Appendix A).

### 4.2. Consensus Map and QTL Projection

A consensus map was generated using the IBM2 2008 Neighbours map as a reference map and by incorporating the markers flanking the QTLs identified in individual studies using the LPmerge package of the R programming [85] as described by Kumar et al. [86]. Two types of text files, namely, QTL information and the genetic map, were prepared for QTL projection using the data provided in individual studies following the instructions given in the manual of BioMercator V4 [87]. The genetic map file contained the following information: mapping function, map units, type and size of the population, and locations of different markers on different linkage groups/chromosomes. The QTL information file had the following points for each QTL: QTL ID (each QTL was assigned a unique identity for analysis), associated root traits, chromosome number, the peak position of QTL, CI (95%), LOD score, and PVE value. 

The individual QTLs were projected onto the consensus map using the QTLProj command of BioMercator v4.2.3 software. Meta-analysis computing was based on the positions of each input QTL and the variance of these positions, which was measured using CIs of individual QTLs. MQTL analysis was performed using the Veyrieras two-step algorithm. Firstly, based on the lowest value of selection criteria, the best MQTL model was chosen in at least three of the models (Bayesian information criterion (BIC), Akaike information criterion (AIC), AIC correction (AICc), AIC model 3 (AIC3), and average weight of evidence (AWE)). In a further step, the selected best model was utilized to ascertain the following: (i) their consensus positions (based on the variance of input QTL positions); (ii) the number of MQTLs on each chromosome (based on the number of input QTLs on the consensus map); and (iii) 95% CI (based on the variance of input QTL intervals) (as described in [30]).The MQTLs predicted during the present study were designated as MQTL2.1, MQTL2.2, and so on. The name MQTL2.1 represents its position and order on the corresponding chromosomes (in “2.1”, “2” represents the chromosome number, and “1” is the order of its position on the corresponding chromosome). The LOD scores and PVE values of individual MQTLs were calculated as the mean values of the LOD scores, and PVE values are the initial QTLs clustered in the MQTL region. 

### 4.3. Determining the Physical Positions of MQTLs and Validation with GWAS

The physical positions of the MQTLs were obtained by browsing the flanking markers’ physical location in the JBrowser of maizeGDB (https://jbrowse.maizegdb.org/ (accessed on 14 June 2022)) by utilizing the latest version of genome assembly (*Zm*-B73-REFERENCE-NAM-5.0). Some markers’ physical locations not available in JBrowser were retrieved directly from the Gramene Marker Database (https://archive.gramene.org/markers/ (accessed on 14 June 2022)). Data on MTAs from 10 GWA studies for root-related traits were retrieved and utilized for checking the efficacy of MQTL regions. The size of the populations or association panels evaluated in these studies varied from 80 to 508 lines/genotypes. The salient features of these studies are listed in Appendix A. Maize has a linkage disequilibrium (LD) of about 350 Kbp [88,89]. Henceforth, the MTAs obtained from GWAS near MQTL in a ±350 Kbp physical region were considered co-located with MQTL.

### 4.4. Establishment of MQTLs for Gene Mining and Selection of Some Promising MQTLs for Breeding

Among the MQTLs predicted during the present study, some of the MQTLs were categorized as hcMQTL (to be utilized for CG mining) and some as bMQTLs (having potential to be utilized in breeding programs). The hcMQTLs were selected based on the following criteria: (i) physical interval less than 1Mb, (ii) each based on more than 10 initial QTLs from different studies. Somewhat different criteria were followed for the identification of bMQTLs [27], as discussed below: (i) mean PVE greater than 10%, (ii) each based on at least 5 initial QTLs, and (iii) CI less than 2cM. The genes models available within hcMQTL regions were retrieved using the BioMart tool of theEnsemblPlants database (http://plants.ensembl.org/index.html (accessed on 25 June 2022)). The following data were retrieved for each gene model: gene ID, gene position (bp), gene ontology (GO) term name, and GO domain. Functional annotation of these gene models was performed based on the domains in the corresponding protein sequences, which were obtained using the InterPro database (https://www.ebi.ac.uk/interpro/ (accessed on 27 June 2022)). We performed GO enrichment analysis on candidate genes associated with root traits using the AgriGO (v2.0) website (http://systemsbiology.cau.edu.cn/agriGOv2/ (accessed on 21 February 2023)). We considered a term to be enriched in the corresponding gene set if its P value was less than 0.001. The results of the GO enrichment analysis were displayed using a “bubble chart”, which was created using an online platform SRPlot (https://www.bioinformatics.com.cn/srplot (accessed on 21 February 2023)).

### 4.5. Expression Analysis of Gene Models

Expression data of the genes present within the hcMQTLs regions from two studies [90,91] were searched using an online expression tool, “qTeller” (https://qteller.maizegdb.org/ (accessed on 29 June 2022)). The transcriptomic data of different root tissues at different time intervals [84] were available for: the primary root after three days of sowing (Primary_Root_3DAS); differentiation, meristematic and elongation zones of the primary root after 3 DAS (Root_DZ_3DAS, Root_MZ_and_EZ_3DAS); root stele and cortical parenchyma after 3 DAS (Root_Stele_3DAS, Root_CP_3DAS); the primary root in the greenhouse after 6 DAS (Primary_Root_GH_6DAS); root system (Root_System_7DAS), primary root (Primary_Root_7DAS), seminal roots (Seminal_Roots_7DAS), and primary root zone 1 after 7 DAS (Primary_Root_Z1_7DAS). Walley et al. [85] showed the expression profiling of different root tissues such as brace root (V13 stage), crown root (V13), primary root seminal root, whole root system, whole primary root, tap root, root cortex, root meristem, and root elongation zone at different time intervals. The gene models with ≥3 transcripts per million (TPM) expressions were considered as significantly expressed in various root tissues. Heat maps of some genes were developed using the Heatmapper tool (http://heatmapper.ca/ (accessed on 29 June 2022)).

### 4.6. Unravelling Conserved Genomic Regions Associated with Root Traits among the Cereals

The syntenic and collinear regions associated with root traits in maize, rice, and sorghum genomes were identified by the following two major steps: (i) identification of orthologs of maize genes residing within MQTL regions in rice and sorghum genomes with BLAST analysis using the “BioMart” tool (http://plants.ensembl.org/biomart/ (accessed on 30 June 2022)) available at the Ensembl Plants database [27], (ii) extraction of physical positions of genes and other required information in rice and sorghum orthologs. The data generated during the current study and from earlier published studies in rice [35,36] were explored further for identifying ortho-MQTLs between maize and rice genomes. For this purpose, the physical positions of the rice orthologues were investigated to see if they were co-localized in the rice MQTL regions, and rice MQTLs harbouring at least ten corresponding genes were considered as ortho-MQTLs of maize root traits. However, due to a lack of MQTL data (i.e., information on MQTLs associated with root traits) in sorghum, ortho-MQTL analysis between maize and sorghum could not be conducted. The syntenic relationships among maize, rice, and sorghum genomes were represented as circos plot using an R-based package R/shiny [92].

### 4.7. MQTL Characterization Using Cloned Genes and Homology of MQTLs with Rice

Several genes associated with root traits have previously been characterized and cloned in maize. The physical positions of the candidate genes were retrieved from MaizeGDB and compared with MQTL regions to determine their localizations within or near the MQTL regions. Further, a bibliographic search was conducted to identify orthologs of known rice genes in MQTL regions. These rice genes are known to play roles in root development and nutrient uptake.

## 5. Conclusions

The current study combined previously identified QTLs with maize genomic and transcriptomic resources to improve the understanding of the genetic architecture of root-related traits in maize by identifying MQTLs, ortho-MQTLs, and CGs. 68 MQTLs were identified, including 36 MQTLs validated with GWAS-MTAs, each with a narrow CI and 68 potential CGs. Among the 68 MQTL regions, nine were considered bMQTLs, and five were validated with GWAS-based MTAs. We recommend using these five bMQTLs in MAS to model RSA in maize. Further, the conserved genes among maize, rice, and sorghum genomes may aid in understanding the molecular basis and evolutionary processes involved in root formation in different cereals. The flanking markers to some of the promising MQTLs can be used in genomic selection models as fixed effects and thus improve the accuracy of genomic prediction for root traits.

## Figures and Tables

**Figure 1 ijms-24-06135-f001:**
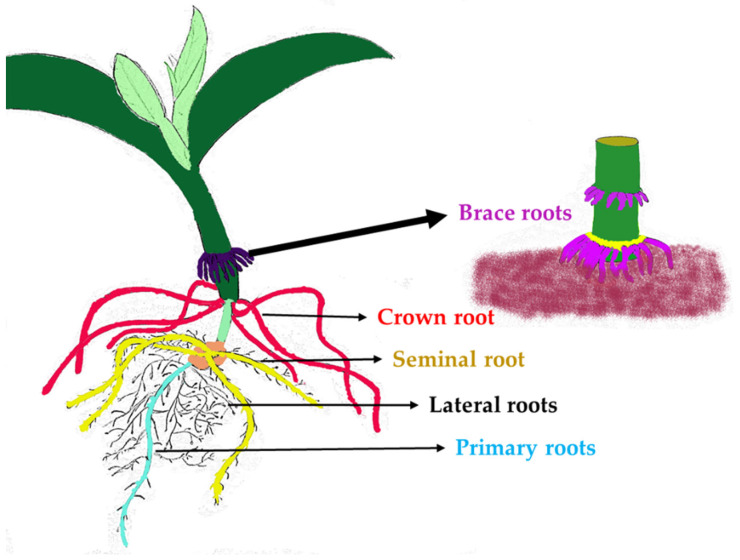
Root system architecture (RSA) of maize. Different types of maize roots are represented in different colours.

**Figure 2 ijms-24-06135-f002:**
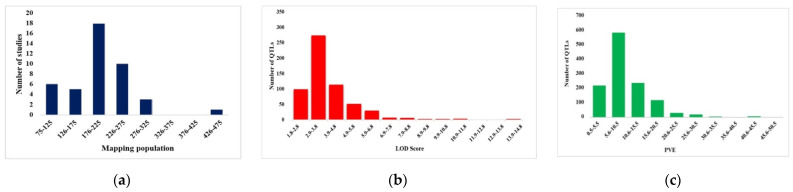
Frequency distribution of (**a**) population size used in the individual studies (blue bar chart); (**b**) LOD score of initial QTLs (red bar chart); (**c**) PVE (%) of initial QTLs (green bar chart) from the previous studies.

**Figure 3 ijms-24-06135-f003:**
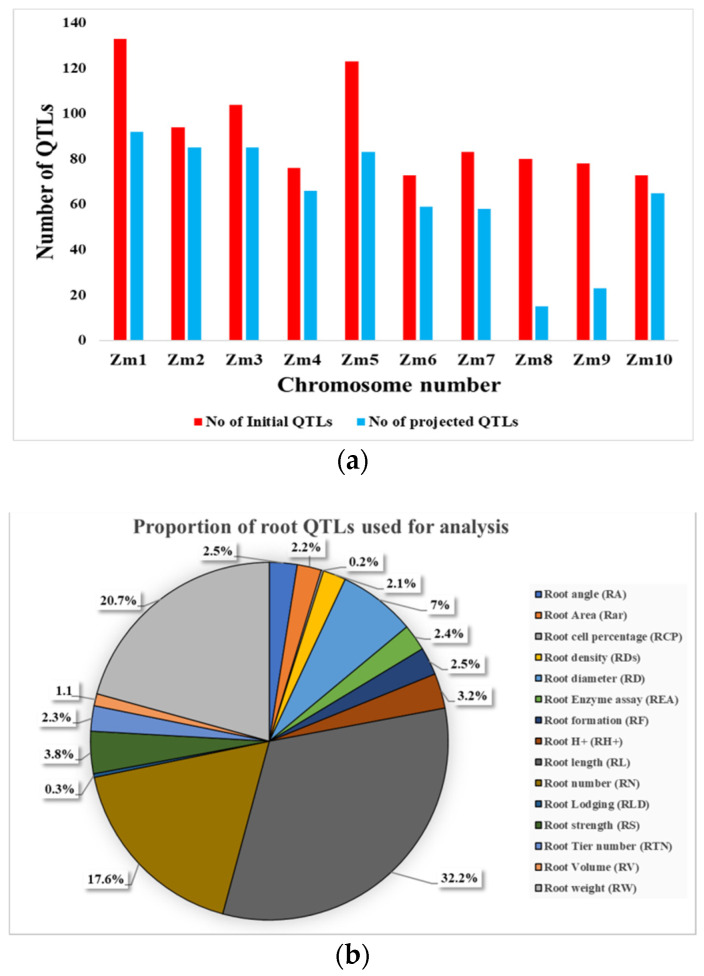
(**a**) Bar graph representing the differences in number of QTLs before (red bar) and after projection (blue bar). (**b**) Pie chart representing proportion of root QTLs in different colours used for projection. The pie chart starts from root angle (RA; 2.5%) and ends with root weight (RW; 20.7%).

**Figure 4 ijms-24-06135-f004:**
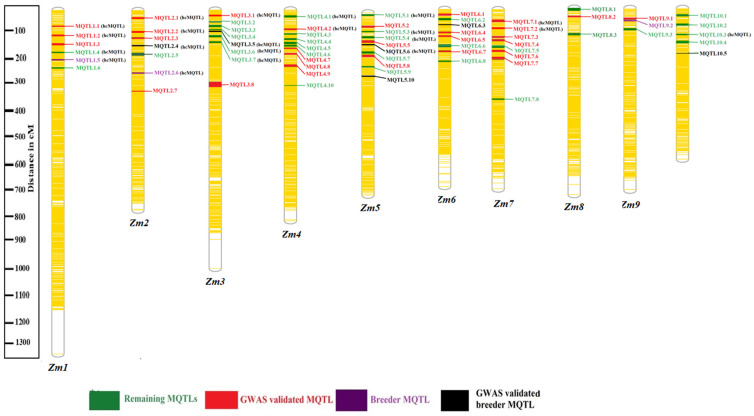
The distribution of different MQTLs on maize chromosomes. The explanation for different colours utilized to represent the MQTL is given at the bottom of the figure.

**Figure 5 ijms-24-06135-f005:**
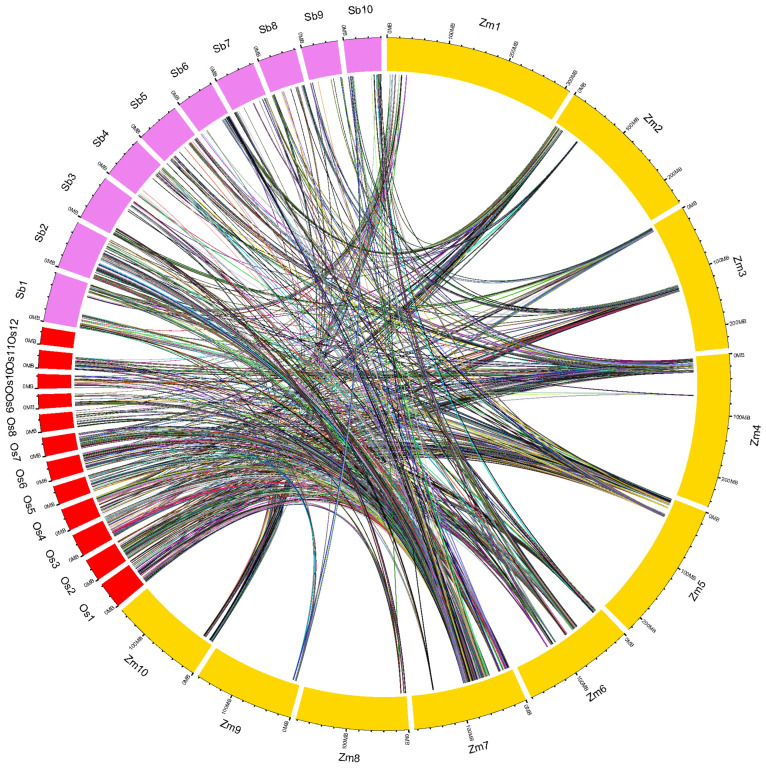
Conserved genomic regions among maize (*Zm*), rice (Os) and sorghum (Sb) genomes. The outer circles represent the maize (yellow), rice (red), and sorghum (violet) chromosomes. The linking lines represent the syntenic regions of maize–rice and maize–sorghum.

**Figure 6 ijms-24-06135-f006:**
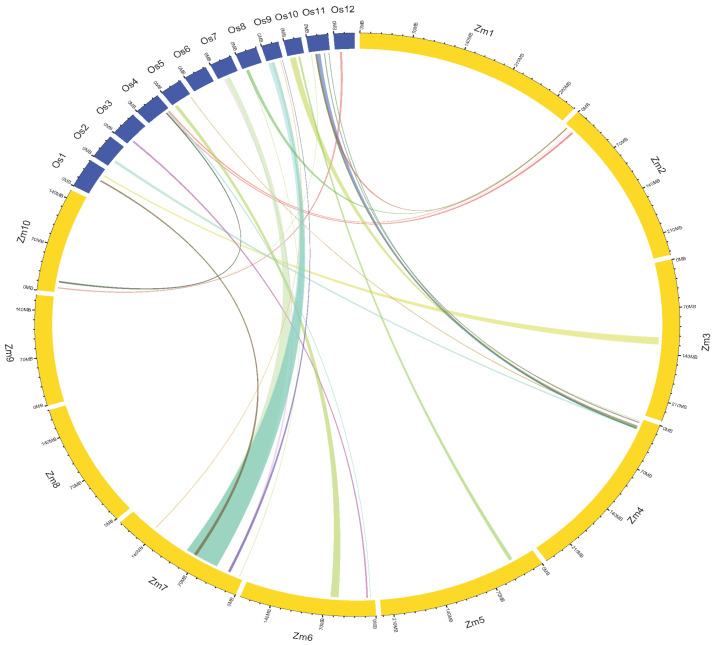
Conserved regions among ortho-MQTLs in maize (*Zm*) and rice genomes (Os). The outer circles represent the maize (yellow) and rice (blue) chromosomes. The linking lines represent the syntenic regions of rice and maize.

**Figure 7 ijms-24-06135-f007:**
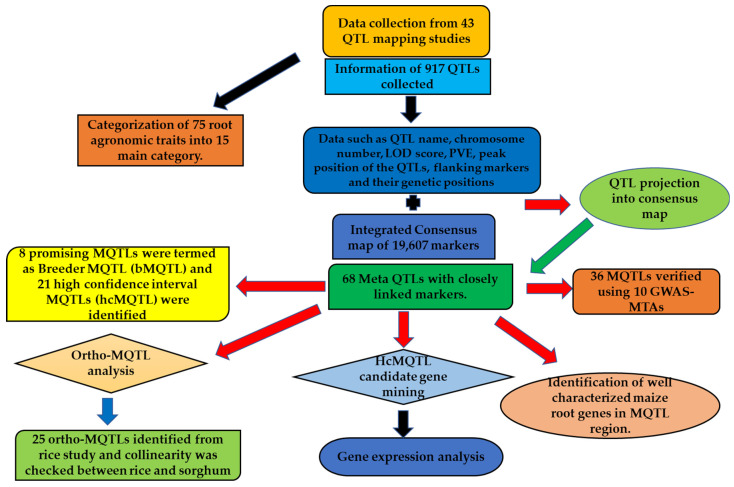
Schematic representation of the steps involved in the meta-QTL analysis conducted during the present study.

**Table 1 ijms-24-06135-t001:** A summary of MQTLs identified during the present study.

S. No	MQTL Name	Chromosome	CI (95%)	Genetic Position (cM)	Flanking Maker (From-to)	Number of QTLs Involved	Root Trait Categories
1	MQTL1.1	1	2.35	60.39–62.74	ofp1-gpm330	42	RA, RD, REA, RL, RN, RS, RVT, and RW
2	MQTL1.2	1	2.55	95.86–98.41	mrpa4-lls1	15	RD, REA, RL, RN, RS, and RW
3	MQTL1.3	1	4.18	128.74–132.92	sdg122-cox10b	5	RD, RL, RN, and RS
4	MQTL1.4	1	2.45	159.71–162.16	TIDP3048-ereb35	13	RA, RDT, RL, RN, RTN, and RW
5	MQTL1.5	1	0.95	187.73–188.68	pco109407-pco107465a	12	RAR, RL, RN, RVT, and RW
6	MQTL1.6	1	2.75	218.80–221.55	pmk1-pco141386	5	RN, RLD, and RW
7	MQTL2.1	2	2.89	27.99–30.88	isu144a-kcbp1	16	RD, RH+, RL, RN, and RW
8	MQTL2.2	2	3.58	80.68–84.26	ereb47-AY109603	14	RA, RD, REA, RH+, RL, RN, and RTN
9	MQTL2.3	2	4.24	103.66–107.9	gpm914a-TIDP3748	7	RDT, RD, RL, and RTN
10	MQTL2.4	2	0.9	134.98–135.88	Bin2-448-mex1	17	RA, RDT, RL, RN, RS, and RW
11	MQTL2.5	2	9.53	162.31–171.84	mybr20-IDP650	9	RDT, RL, and RW
12	MQTL2.6	2	1.79	237.83–239.62	TIDP8875-les1	13	RAR, RL, RN, and RW
13	MQTL2.7	2	1.19	307.99–309.18	abi17-TIDP3679	9	RD, RF, RL, RN, and RW
14	MQTL3.1	3	2.54	19.79–22.33	agrr209a-Bin3_33	21	RAR, RD, RL, RN, RS, RTN, RVT, and RW
15	MQTL3.2	3	3.89	42.97–46.86	mmp38-bcd734b	5	RDT, RL, RN, and RW
16	MQTL3.3	3	2.95	59.20–62.15	asg30c-cdo345b	5	RAR, RH+, and RL
17	MQTL3.4	3	1.72	73.56–75.28	bnlg1325-mHbrMC376-Mo17	7	RD, REA, RL, RN, and RW
18	MQTL3.5	3	1.61	79.90–81.51	umc1886-cr1	19	RH+ RL, RN, and RW
19	MQTL3.6	3	5.21	97.57–102.78	mHbrBA221_Mo17-plpb1	11	RD, REA, RH+, RL, RN, and RW
20	MQTL3.7	3	2.43	122.41–124.84	SC298C-lim66	15	RA, RDT, RL, RN, and RW
21	MQTL3.8	3	18.56	273.95–292.51	bhlh58-TIDP3495	2	RLD, andRW
22	MQTL4.1	4	4.88	22.82–27.7	IDP8455-UMC156A	16	RDT, RL, RN, RS, and RW
23	MQTL4.2	4	3.2	72.47–75.67	IDP483-bnlg1318	12	RD, RL, RN, RS, and RW
24	MQTL4.3	4	2.71	90.76–93.47	uaz68b(zp19)-Bin4_384	5	RL, RN, and RW
25	MQTL4.4	4	4.27	108.91–113.18	PZE_108065982-GRM*ZM*2G157241	6	RDT, RD, REA, and RL
26	MQTL4.5	4	4.84	124.66–129.5	phd28-zhd1	2	RL, and RW
27	MQTL4.6	4	3.72	134.19–137.91	alf18-bnlg292	6	RL, and RN
28	MQTL4.7	4	2.18	146.18–148.36	gpm147-IDP6896	4	RL, RN, and RW
29	MQTL4.8	4	2.66	167.26–169.92	PZE_105145239-gpm782	3	RDT, RL, and RW
30	MQTL4.9	4	6.04	210.71–216.75	TIDP2704-nbcs12	6	RAR, RD, RL, RN, and RW
31	MQTL4.10	4	0.51	287.60–288.11	gpm420a-umc42a	6	RA, RAR, RD, RL, and RW
32	MQTL5.1	5	3.15	7.38–10.53	CDO87A-AY110625	13	RAR, RD, RH+, RL, RVT, and RW
33	MQTL5.2	5	5.47	50.43–55.90	umc1901-chi3	5	RL
34	MQTL5.3	5	3.08	71.28–74.36	umc2388-bnl7.21c	15	RDT, RH+, RL, RN, and RW
35	MQTL5.4	5	5.68	90.85–96.53	cdpk18-rab28	13	RD, REA, RH+, RL, and RW
36	MQTL5.5	5	2.7	112.13–114.83	gpm921b-PGAMCGG330	5	RAR, RN, and RTN
37	MQTL5.6	5	1.5	120.06–121.56	orc5-SYN37265	14	RDT, RD, RL, RN, and RW
38	MQTL5.7	5	5.02	148.31–153.33	agrx43-IDP8129	3	RL, and RN
39	MQTL5.8	5	2.71	163.09–165.80	umc107b-bnlg1208	8	RA, RAR, RL, and RN
40	MQTL5.9	5	2.12	205.46–207.58	ca2p10-smh6	2	RDT, and RVT
41	MQTL5.10	5	1.97	240.76–242.73	IDP5960-bnlg1700	5	RD, RL, and RN
42	MQTL6.1	6	4.8	17.05–21.85	AX_86316369-umc1572	12	RDT, RD, RL, RN, RS, and RW
43	MQTL6.2	6	4.66	35.11–39.77	umc2698-si486014c02	9	RAR, RH+, RL, RN, and RW
44	MQTL6.3	6	0.97	57.64–58.61	lht1-gpm304a	12	RD, RH+, RL, RN, RS, and RW
45	MQTL6.4	6	3.47	86.335–89.805	ago1a-magi84474	13	RA, RCP, RD, RL, RN, RTN, and RW
46	MQTL6.5	6	3.85	101.36–105.21	csu680e-abi29	4	RL, and RW
47	MQTL6.6	6	8.01	134.25–142.26	IDP350-mHbrBG152-B73	4	RDT, RL, and RW
48	MQTL6.7	6	5.36	157.84–163.2	SYN1223-c3h26	2	RL
49	MQTL6.8	6	1.18	197.14–198.32	IDP8221-pco062823a	3	RA, and RL
50	MQTL7.1	7	4.22	40.74–44.96	phi057-umc1112	20	RD, RL, RN, RS, RVT, and RW
51	MQTL7.2	7	4.07	68.59–72.66	IDP1692-PGAMCGG275	14	RD, REA, RL, RN, RS, and RW
52	MQTL7.3	7	3.87	102.66–106.53	csu486b-npi400a	8	RA, RL, RN, RTN, and RW
53	MQTL7.4	7	3.84	114.17–118.01	umc1888-umc2190	3	RA, RL, and RN
54	MQTL7.5	7	3.92	140.84–144.76	v5-pzb01228	4	RD, RL, and RN
55	MQTL7.6	7	5.43	155.27–160.70	gpm273b-sid1	4	RA, RD, RL, and RN
56	MQTL7.7	7	7.76	180.84–188.6	arftf37-sdg110a	2	RW
57	MQTL7.8	7	2.02	340.05–342.07	pebp24-gpm369c	3	RL
58	MQTL8.1	8	10.64	0–10.36	ms23-umc1786	3	RL, and RN
59	MQTL8.2	8	6.79	27.56–34.35	psei4-IDP1987	6	RF, RH+, RL, and RN
60	MQTL8.3	8	8.49	95.06–103.55	gpm932d-IDP5056	6	RF, RL, and RW
61	MQTL9.1	9	2.13	36.17–38.30	pip1f-mads74	11	RD, RL, RN, and RW
62	MQTL9.2	9	1.31	44.93–46.24	opr1-lim343	7	RL, and RW
63	MQTL9.3	9	5.73	76.43–82.16	IDP1969-agrr147	5	RL, RN, and RW
64	MQTL10.1	10	3.24	25.84–29.08	myb69-ksu1e	7	RA, RLT, and RW
65	MQTL10.2	10	5.79	59.40–65.19	TIDP3129-phm3631	8	RA, RD, RL, RN, and RS
66	MQTL10.3	10	2.16	99.34–101.5	umc1053-dmag4	28	RAR, RD, REA, RL, RN, RS, RTN, and RW
67	MQTL10.4	10	5.39	126.06–131.45	SYN22956-umc2749	9	RD, RL, RN, RLD, and RS
68	MQTL10.5	10	0.62	171.74–172.36	IDP4076-de18	13	RF, RH+, RL, RTN, and RW

RA—root angle, RAR—root area, RCP—root cell percentage, RD—root density, RD—root diameter, REA—root enzyme assay, RF—root formation, RH+—root H+, RL—root length, RN—root number, RLD—root lodging, RS—root strength, RTN—root tier number, RV—root volume, RW—root weight.

**Table 2 ijms-24-06135-t002:** Known maize genes co-localizing with different MQTLs identified during the present study.

Known Genes Found within the MQTL Regions									
S. No	Gene Name	MQTL Name	Gene Stable ID	Chromosome	GENE Start (bp)	Gene End (bp)	Role	Gene Product	Reference
1	*PHT1*	MQTL 5.1	*Zm*00001eb222510	5	33,267,448	33,269,832	Phosphate utilization activity	Phosphate permease	[38]
2	*YUC3*	MQTL 4.7	*Zm*00001eb168680	4	15,140,763	15,143,517	Auxin biosynthesis in roots	Flavin monooxygenase	[39]
3	*YUC6*	MQTL 8.2	*Zm*00001eb332870	8	3,246,199	3,249,205	Auxin biosynthesis in roots	Flavin monooxygenase	[39]
4	*DXS2*	MQTL 7.5	*Zm*00001eb302370	7	14,390,493	14,393,714	Highly expressed in the primary root of young seedlings and complete function not known yet	Transketolase, *N*-terminal	[40]
5	*TUA4*	MQTL 5.7	*Zm*00001eb215710	5	9,999,369	10,002,552	Structural maintenance and cell division in root cells	alpha Tubulin	[41]
6	*RTP1*	MQTL 4.6	*Zm*00001eb168290	4	13,009,615	13,011,188	Highly expressed in roots of young seedlings; involved in suberin pathway	O-methyltransferase domain	[42]
**Known Genes Found near the MQTL Regions (±1 Mb MQTL Regions)**									
1	*RTCS1*	MQTL1.2	*Zm*00001eb003920	1	10,913,046	10,913,187	Promotes primary root growth in plant	Rootless concerning crown and seminal roots1	[43]
2	*NSA1*	MQTL2.4	*Zm*00001eb071820	2	12,627,413	12,631,767	Involved in Na+ homeostasis; mutants of the gene promote root Na+ efflux	Na+ content under saline-alkaline conditions1	[44]
3	*EIN2*	MQTL3.1	*Zm*00001eb119690	3	2,916,990	2,926,153	Produces secondary metabolites in roots under stress	Ethylene insensitive 2	[45]
4	*NHX1*	MQTL4.2	*Zm*00001eb165870	4	4,724,901	4,733,653	Involved in Na+ homeostasis; mutants promote root Na+ efflux	Na+/H+ antiporter 1	[46]
5	*CPPS4*	MQTL4.3	*Zm*00001eb167120	4	7,053,803	7,059,057	Upregulates under root exposure to abiotic stress	Copalyl diphosphate synthase4	[47]
6	*BIGE1*	MQTL5.1	*Zm*00001eb211050	5	1,856,121	1,860,385	Mutant causes accelerated root initiation as well as enlargement of the embryo scutellum	big embryo1	[48]
7	*RCP1*	MQTL6.5	*Zm*00001eb270090	6	78,255,812	78,258,499	Highly expressed in outer most root cap cells and involved in root development	Root cap protein1	[49]
8	*SKUS13*	MQTL7.1	*Zm*00001eb298980	7	2,650,096	2,655,635	Involved in directional root growth	Skewed root growth similar 13	[50]
9	*YUC5*	MQTL7.5	*Zm*00001eb306140	7	39,946,780	39,950,489	Transcribed in the root apex (0–5mm)	Yucca5: flavin monooxygenase-like enzyme	[51]
10	*AW330564*	MQTL10.1	*Zm*00001eb405820	10	2,736,099	2,736,268	Root phototrophism	Similar to M. truncatula Root phototropism protein	[52]

**Table 3 ijms-24-06135-t003:** Rice genes and their orthologs within MQTL regions associated with different root traits.

Known Rice Genes	Rice Gene Stable ID	Rice Chromosome	Encoded Proteins	Function Description	Corresponding Maize Gene	Maize MQTL	Reference
*OsACS7*	*Os07g0280200*	7	ACC synthase	Involved in phosphate uptake in plants	*Zm00001eb308860*	MQTL7.7	[53]
*OsAMT1*	*Os04g0509600*	4	Ammonium transporter	Ammonium ion uptake in roots	*Zm00001eb077350*	MQTL2.6	[54]
*OsCDPK10*	*Os03g0788500*	3	Calcium-dependent protein kinase	Signalling of various root related biological process	*Zm00001eb213280*	MQTL5.4	[55]
*OsCIPK14*	*Os12g0113500*	12	Serine/threonine protein kinase	Signalling of various root related biological process	*Zm00001eb406220*	MQTL10.2	[56]
*OsCIPK15*	*Os11g0113700*	11	Serine/threonine protein kinase	Signalling of various root related biological process	*Zm00001eb406220*	MQTL10.3	[56]
*OsCIPK23*	*Os07g0150700*	7	Serine/threonine protein kinase	Signalling of various root related biological process	*Zm00001eb300010*	MQTL7.2	[56]
*OsCKX1*	*Os01g0187600*	1	Cytokinin Oxidase	Cytokinin metabolism	*Zm00001eb121500*	MQTL3.6	[57]
*OsEXPA10*	*Os04g0583500*	4	Expansin	Root cell elongation	*Zm00001eb072890*	MQTL2.5	[58]
*OsHAK10*	*Os06g0625900*	6	Potassium transporter	K+ transport from soil to plants	*Zm00001eb275630*	MQTL6.7	[59]
*OsLAC17*	*Os10g0346300*	10	Laccase	Catalyses the synthesis of lignin under stress	*Zm00001eb165090*	MQTL4.1	[60]
*OsMGT*	*Os06g0650800*	6	Magnesium transporters	Magnesiumion uptake and transport from root to other organs	*Zm00001eb274990*	MQTL6.7	[61]
*OsNAC7*	*Os06g0131700*	6	NAC TF	Modulates root growth	*Zm00001eb269810*	MQTL6.5	[62]
*OsWRKY68*	*Os04g0605100*	4	*WRKY* TF	Involved in phosphate uptake in plants	*Zm00001eb071590*	MQTL2.4	[63]
*OsYUCCA10*	*Os01g0274100*	1	Flavin monooxygenase-like enzyme	Auxin biosynthesis	*Zm00001eb332870*	MQTL8.2	[64]
*OsYUCCA12*	*Os02g0272200*	2	Flavin monooxygenase-like enzyme	Auxin biosynthesis	*Zm00001eb168680*	MQTL4.7	[64]
*OsYUCCA13*	*Os11g0207700*	11	Flavin monooxygenase-like enzyme	Auxin biosynthesis	*Zm00001eb168680*	MQTL4.7	[64]
*OsYUCCA14*	*Os11g0207900*	11	Flavin monooxygenase-like enzyme	Auxin biosynthesis	*Zm00001eb168680*	MQTL4.7	[64]
*OsYUCCA9*	*Os01g0273800*	1	Flavin monooxygenase-like enzyme	Auxin biosynthesis	*Zm00001eb332870*	MQTL8.2	[64]
*OsDREB1E*	*Os04g0572400*	4	*DREB1* TF	Upregulated in roots during stress	*Zm00001eb073550*	MQTL2.5	[65]

## Data Availability

Data generated or analysed during this study are included in this published article (and its Appendix A).

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
