# Peer review of "Genome-Wide Meta-Analysis of QTLs Associated with Root Traits and Implications for Maize Breeding"

_ijms, 2023, doi:10.3390/ijms24076135_

Round 1

Reviewer 1 Report

Thank you for the opportunity to review the manuscript titled “Genome-wide meta-analysis of QTLs associated with root traits and implications for maize breeding”. The MS reports a meta-QTL analysis of root system architecture (RSA) for maize, an important trait for plant growth and stress responses. The authors have tried to incorporate 917 QTL from multiple individual QTL/GWAS mapping studies to build 68 MQTL blocks. The outcome of the analysis would be useful not only for maize research community but also for maize breeders to prioritise which QTL to incorporate into their breeding programs. The paper’s ideas were clear, it was well written, therefore it was so easy to read. I have a few minor comments for the authors to address:

1.      Figure 3 need to be fixed, A and B need to be swapped

2.     FigureS2 was not included in the supplementary documents

3.     I don’t know what would be the meaning of expression data here?? Some explanation for that would be great. Do you mean they are functional?

4.     Clearer explanation on prediction of genes associated with RSA within hcMQTL regions would be needed.

5.     Some English expressions or typos as below

a.     Line 64, P2: Figure 1, “Different types maize roots” changes to “Different types of maize roots”

b.     Line 67-68, P3: add numbers of QTL studies used for this paper instead of “mapping studies published from 1998 to 87 2020” to be compatible to “20” studies used by previous MQTL study for maize root.

c.     Line 101, P3: “The LOD score of each QTL ranged” pls change to “The LOD score of these QTL ranged”

d.     Line 134, P4: “the maximum projected QTLs (92) 134 were on Zm10” is this correct? Pls check.

e.     Line 188, P9: Why “69 were selected and recommended”?

f.      Line 201, P9: what was the analysis and how the authors get to this gen number “Sorghum and maize had 2366”?

g.     Line 226, P11: pls replace “some” with a number

h.     Line 390, P16 - Table 3: missing space “their orthologswithin”

i.      Line 444, P18: missing ‘)’ in “(as described in [30]”

j.      Line 466-467, P19: Missing ‘(‘ in “in breeding programs)”

k.     Line 470,496,480; P19: pls remove ‘-‘ and replaced with ‘:’

l.      Line 524-525, P20: did you really show “evolutionary processes involved”???

Author Response

Query 1:      Figure 3 need to be fixed, A and B need to be swapped

Response 1: The Figure 3 has been fixed.

Query 2:     Figure S2 was not included in the supplementary documents

Response 2: Figure S2 has been included in the supplementary documents. 

Query 3:     I don’t know what would be the meaning of expression data here?? Some explanation for that would be great. Do you mean they are functional?

Response 3: Expression data means the transcript profile of the genes available in the maize database which has been analyzed to identify candidate genes present within and near the MQTLs related to root-traits. Actually, we used some previously published expression datasets to determine whether the genes discovered during the current study have been found to be significantly expressed in any root tissues. Our hypothesis was that a gene would be involved in the regulation of root-related traits if it exhibits significant expression in the root tissues. It has been described in section 4.5.

Based on the expression pattern they are found to be functional.

Query 4:    Clearer explanation on prediction of genes associated with RSA within hcMQTL regions would be needed.

Response 4: The hcMQTLs were selected based on the following criteria: (i) physical interval less than 1Mb, (ii) each based on more than 10 initial QTLs from different studies. These selected hcMQTLs were considered for CG mining. The genes models available within hcMQTL regions were retrieved using the BioMart tool of EnsemblPlants database (http://plants.ensembl.org/index.html). The following data was retrieved for each gene model: gene ID, gene position (bp), gene ontology (GO) term name, and GO domain. Functional annotation of these gene models performed based on the domains in the corresponding protein sequences, which were obtained using the InterPro database (https://www.ebi.ac.uk/interpro/). The revised manuscript now includes detailed information on predicting candidate genes from hcMQTL regions in material and methods section.

Query 5:     Some English expressions or typos as below

  1. Line 64, P2: Figure 1, “Different types maize roots” changes to “Different types of maize roots”

Response: Needful is done.

  1. Line 67-68, P3: add numbers of QTL studies used for this paper instead of “mapping studies published from 1998 to 87 2020” to be compatible to “20” studies used by previous MQTL study for maize root.

Response: 43 mapping studies has been added.

  1. Line 101, P3: “The LOD score of each QTL ranged” pls change to “The LOD score of these QTL ranged”

Response: The necessary changes have been made.

  1. Line 134, P4: “the maximum projected QTLs (92) 134 were on Zm10” is this correct? Pls check.

Response: It is typo error, correct statement is ‘the maximum projected QTLs (92) were on Zm1’. This has been rectified in revised manuscript.

  1. Line 188, P9: Why “69 were selected and recommended”?

Response: As many as 235 gene models were found to be significantly expressed in different root tissues. Based on the available literature, the functional significance of the proteins for root traits encoded by these genes was examined. This analysis identified 69 genes that encode proteins that are functionally important for the regulation of RSA, and as a result, these genes were recommended as potential candidate genes for the corresponding hcMQTL regions. These details have now been provided in the revised manuscript.

  1. Line 201, P9: what was the analysis and how the authors get to this gene number “Sorghum and maize had 2366”?

Response: The syntenic and collinear regions associated with root traits in maize, rice, and sorghum genomes were identified by following two major steps as given below: (i) identification of orthologs of maize genes residing within MQTLs in rice and sorghum genomes using BLAST analysis, (ii) extraction of physical positions and other required information in rice and sorghum orthologs. These details have been provided in the 4.6 section of the revised manuscript.

  1. Line 226, P11: pls replace “some” with a number

Response: Corrected as suggested.

  1. Line 390, P16 - Table 3: missing space “their orthologs within”

Response: This has been corrected as suggested.

  1. Line 444, P18: missing ‘)’ in “(as described in [30]”

Response: The needful is done.

  1. Line 466-467, P19: Missing ‘(‘ in “in breeding programs)”

Response: The needful is done.

  1. Line 470,496,480; P19: pls remove ‘-‘ and replaced with ‘:’

Response: The needful is done.

  1. Line 524-525, P20: did you really show “evolutionary processes involved”???

Response: In the current study, we discovered a number of conserved genes in the genomes of maize, rice, and sorghum that may help us understand the molecular basis and evolutionary processes underlying the development of roots in different cereals. In the present study, we did not investigate the evolutionary processes; rather, we provided some future directions for evolutionary research.

 All the comments/suggestions by reviewer 1 are highlighted in yellow color.

Reviewer 2 Report

In this manuscript, Karnatam et al. performed a meta-QTL analysis of 43 studies on maize root system architecture (RSA). They discovered two dozen high-confidence QTLs that are also conserved in rice/sorghum. The authors also provided a list of candidate genes for future analysis. Overall, the manuscript is imaginative and provides new insights into maize RSA. I have a few minor suggestions below.

1.    Please provide an introduction to LOD and PVE for readers without a quantitative genetics background.

2.    The legends of Figure 3 a and b are reversed.

3.    I suggest changing the pie chart to a horizontal bar chart. The colors in the pie chart in Figure 3 are very hard to distinguish.

4.    I suggest the authors move Figure 7 to Figure 1 so that the readers get the pipeline at the beginning.

5.    Please provide a reference for the Biomercator software on lines 425 and 433.

6.    On line 496, please provide the details of the ortholog finding. What BLAST settings and cutoffs were used?

7.    Please provide details about the synteny mapping between species.

Author Response

Query 1.    Please provide an introduction to LOD and PVE for readers without a quantitative genetics background.

Response 1. These terms are common for a breeder or any geneticist so it needs no introduction.

Query 2.    The legends of Figure 3 a and b are reversed.

Response 2. The needful changes have been done.

Query 3.    I suggest changing the pie chart to a horizontal bar chart. The colors in the pie chart in Figure 3 are very hard to distinguish.

Response 3. We have updated the pie chart with more clarity and also added percentages of each trait in pie chart.

Query  4.    I suggest the authors move Figure 7 to Figure 1 so that the readers get the pipeline at the beginning.

Response 4. Respectfully, this figure 7 is best suited in Material and methods section.

Query 5.    Please provide a reference for the Biomercator software on lines 425 and 433.

Response 5. The reference has been cited [85] and listed in reference section of the revised manuscript.

Query 6.    On line 496, please provide the details of the ortholog finding. What BLAST settings and cutoffs were used?

Response 6. For this, BLAST analysis against rice and sorghum genome was done using the tool available at Ensembl Plants to identify corresponding orthologues in chromosomes [27]. Ensembl Plants database uses an algorithm for identifying the orthologues. So, cutoffs and BLAST settings are not publicly available.  Ensembl Plants includes synteny analyses, based on pairwise genome alignments, and some generated by our collaborators Gramene.

Query 7.    Please provide details about the synteny mapping between species.

Response 7. Synteny mapping and ortholog finding are synonymous terms. The details of synteny mapping/ ortholog finding have been already provided in the material and method section 4.6. Also, in result section 2.5, the details on synteny mapping have been provided.

All the comments/suggestions by reviewer 2 are highlighted in blue color.

Reviewer 3 Report

This research article entitled ” Genome-wide meta-analysis of QTLs associated with root traits and implications for maize breeding ” provides very useful information and resource associated with root traits. As far as I know, it is an original contribution to our understanding of root system architecture in different crop contexts. In this manuscript, the author started a meta-QTL analysis linked with root traits using 917 QTLs from 43 mapping studies, resulting in 68 MQTLs in maize. Interestingly, more than half of them are consistent with previous studies for root traits. Subsequently, the author performed comparative genomics approaches among maize, rice, and sorghum, confirming some conserved genomic regions related to root traits. Functional and expression analysis revealed many well-known root traits-related genes are located near the MQTLs and many other promising candidate genes are also selected. Overall, this is a very well-written and organized manuscript and provides valuable candidate genes associated with root traits by meta-QTL analysis. Thus, I strongly recommend this manuscript be published in IJMS Journal.

Some suggestions and concerns:

1.       Some of gene names in the manuscript are lowercases, which would be better to use uppercases.

2.       In figure 2, two rows of three bar charts are shown now. Could the author narrow the width of the bar charts and place them in the same row?

3.       In figure 3,  A and B are not in the correct order. Please also add the exact percentages in the pie chart.

4.       In figure 4, the position of different MQTLs seems to locate on the top arms of chromosomes, but not distributed throughout the chromosomes (e.g. Guo et al., 2018, Euphytica; Wang et al., 2022, BMC Plant Biology). Are these novel patterns of the MQTLs regarding the root traits in maize? The authors need to provide more description and add the distance bar (in Mb) in the figure.

5.       As for result 2.4, whether most of the candidate genes on the MQTLs from GO analysis and expression data are associated with the root traits is not clear. Would it be possible to generate the bubble plots to describe GO analysis in more detail?

6.       The current manuscript identified 68 MQTLs, and the previous study by Guo et al. has identified 53 MQTLs (lines). Are there many overlaps or not? Please add some comparisons in the discussion part.

Author Response

Query 1:       Some of gene names in the manuscript are lowercases, which would be better to use uppercases.

Response 1: The gene names have been corrected in the revised manuscript.

Query 2:       In figure 2, two rows of three bar charts are shown now. Could the author narrow the width of the bar charts and place them in the same row?

Response 2: Figure 2 has now been revised as per your suggestion.

Query 3:       In figure 3,  A and B are not in the correct order. Please also add the exact percentages in the pie chart.

Response 3: Figure 3 has now been corrected as desired.

Query 4:       In figure 4, the position of different MQTLs seems to locate on the top arms of chromosomes, but not distributed throughout the chromosomes (e.g. Guo et al., 2018, Euphytica; Wang et al., 2022, BMC Plant Biology). Are these novel patterns of the MQTLs regarding the root traits in maize? The authors need to provide more description and add the distance bar (in Mb) in the figure.

Response 4: Thank you very much for the comment. We agree that the majority of MQTLs are clustered at the fore-end of chromosomes, which are undoubtedly gene-rich regions (https://www.maizegdb.org/genome/assembly/Zm-B73-REFERENCE-NAM-5.0). There are a few reasons why we did not find MQTLs distributed across the chromosomes-

  • The marker density was significantly higher at the fore end of the chromosome than at the end. This is primarily due to the fact that the independent genetic map used to construct the consensus map contained a variety of marker numbers and types. Almost all MQTL studies show this general trend. As a result, most of the MQTLs were found to be condensed at the fore ends of the chromosomes. However, when the physical locations of the markers were taken into account, these MQTLs were found to be located up to 145 Mb regions of the chromosomes, which is comparable to the physical positions of the MQTLs found in the earlier studies.
  • There were some initial QTLs located at the lower ends of the genetic maps of the chromosomes, but they could not be successfully projected (may be because of lack of common markers between initial genetic map and consensus map).
  • The most recent reference genome assembly (Zm-B73-REFERENCE-NAM-5.0) was used in the current study as opposed to the old reference genome assembly (Maize_B73_RefGen_v4) that was used in the earlier studies. The physical positions of markers may have been altered due to differences in genome assemblies used.

Further, we have added a distance bar to the figure as per your suggestion. This distance bar was created to better represent the genetic lengths of the consensus chromosomal maps rather than the physical positions of the chromosomes.

Query 5:      As for result 2.4, whether most of the candidate genes on the MQTLs from GO analysis and expression data are associated with the root traits is not clear. Would it be possible to generate the bubble plots to describe GO analysis in more detail?

Response 5: Thank you very much for your suggestion. As suggested, we performed GO enrichment analysis on candidate genes associated with root traits using the AgriGO (v2.0) website (http://systemsbiology.cau.edu.cn/agriGOv2/). We considered a term to be enriched in the corresponding gene set if its P-value was less than 0.001. The results of the GO enrichment analysis were displayed using a “bubble chart”, which was created using an online platform SRPlot (https://www.bioinformatics.com.cn/srplot” (please have a look at Figure S1 of the revised manuscript). Details are provided in the revised manuscript.

Query 6:       The current manuscript identified 68 MQTLs, and the previous study by Guo et al. has identified 53 MQTLs (lines). Are there many overlaps or not? Please add some comparisons in the discussion part.

Response 6: Thank you very much for your suggestion. As suggested, we compared the physical positions of MQTLs discovered in this study to those discovered in an earlier study by Guo et al (2018). Findings have been discussed in the revised manuscript. A brief summary of this analysis is also given below for your kind consideration-

As many as 22 MQTLs were found to overlap with 23 MQTLs found in the earlier study, according to a comparison of the physical coordinates of MQTLs found in the current study and those found in earlier research by Guo et al. [33]. However, since different genome assembly versions were used to extract the physical coordinates of MQTLs, this comparison might not be entirely accurate. The most recent reference genome assembly (Zm-B73-REFERENCE-NAM-5.0) was used in the current study as opposed to the old reference genome assembly (Maize_B73_RefGen_v4) that was used in the previous study [33]. Additionally, compared to the earlier study [33], more than twice as many QTLs and mapping studies were taken into account in the current study. This could be related to the precise refinement of the physical coordinates of MQTLs predicted in the current study.

All the comments/suggestions by reviewer 2 are highlighted in green color.
